# Hemodialysis and Peritoneal Dialysis in Germany from a Health Economic View—A Propensity Score Matched Analysis

**DOI:** 10.3390/ijerph192114007

**Published:** 2022-10-27

**Authors:** Arim Shukri, Thomas Mettang, Benjamin Scheckel, Isabell Schellartz, Dusan Simic, Nadine Scholten, Martin Müller, Stephanie Stock

**Affiliations:** 1Institute for Health Economics and Clinical Epidemiology (IGKE), Faculty of Medicine, University Hospital Cologne, University of Cologne, 50923 Cologne, Germany; 2Kidney Center Wiesbaden, 65189 Wiesbaden, Germany; 3Institute of Health Care Research, Rhineland State Council, 51109 Cologne, Germany; 4Institute of Medical Sociology, Health Services Research and Rehabilitation Science (IMVR), Faculty of Human Sciences and Medicine, University of Cologne, 50923 Cologne, Germany; 5Department of Emergency Medicine, Inselspital, University Hospital, University of Bern, 3012 Bern, Switzerland

**Keywords:** claims data, costs, ESRD, hemodialysis, peritoneal dialysis, propensity score, German statutory health insurance

## Abstract

Background: Hemodialysis (HD) and peritoneal dialysis (PD) are deemed medically equivalent for therapy of end-stage renal disease (ESRD) and reimbursed by the German statutory health insurance (SHI). However, although the home dialysis modality PD is associated with higher patient autonomy than HD, for unknown reasons, PD uptake is low in Germany. Hence, we compared HD with PD regarding health economic outcomes, particularly costs, as potentially relevant factors for the predominance of HD. Methods: Claims data from two German health insurance funds were analysed in a retrospective cohort study regarding the prevalence of HD and PD in 2013–2016. Propensity score matching created comparable HD and PD groups (*n* = 436 each). Direct annual health care costs were compared. A sensitivity analysis included a comparison of different matching techniques and consideration of transportation costs. Additionally, hospitalisation and survival were investigated using Poisson regression and Kaplan-Meier curves. Results: Total direct annual average costs were higher for HD (€47,501) than for PD (€46,235), but not significantly (*p* = 0.557). The additional consideration of transportation costs revealed an annual cost advantage of €7000 for PD. HD and PD differed non-significantly in terms of hospitalisation and survival rates (*p* = 0.610/*p* = 0.207). Conclusions: PD has a slight non-significant cost advantage over HD, especially when considering transportation costs.

## 1. Introduction

End-stage renal disease (ESRD) is defined as the terminal, persistent stage of chronic kidney disease (CKD) in which a gradual loss of function leads to kidney failure. ESRD is associated with reduced quality of life, premature mortality and high costs for the healthcare system. In fact it is one of the most expensive chronic diseases in Germany [1].

Patients with ESRD need to be monitored carefully, as early referral to either a renal transplant program or dialysis treatment is vital for these patients. There are two different types of dialysis: hemodialysis (HD) and peritoneal dialysis (PD) [2,3]. PD can be performed by the patient at home or in any clean environment, while HD is usually conducted in outpatient dialysis centres three times a week. In retrospective cohort studies, both dialysis modalities are considered equivalent regarding medical outcomes [4,5], and it has been reported that patients on PD report a higher level of autonomy and better health-related quality of life outcomes [6,7,8,9,10,11].

However, at national and regional levels, worldwide PD rates vary widely: e.g., while about four out of five dialysis patients in Hong Kong receive PD (79%) and almost one in four dialysis patients in Sweden (24%) and in the UK (22%), the proportion is only about 8% in the US and 7% in Germany and even only 1% in Luxembourg [12,13]. PD rates depend on patient- and physician-related factors, and on the socioeconomic conditions of the health care system in the different countries [13,14,15,16]. Jain et al. found that 59% of PD patients were from developing countries (i.e., 41% from developed countries), a set of countries that account for an estimated 80% of the global population, with the use of PD increasing in developing countries. PD could have advantages to HD in developing countries, including its simple handling, lower need for trained medical staff, and little need for technical support and conditions [13].

In fact, in Germany both dialysis modalities are remunerated equally by the statutory health insurance (SHI), covering the transport to in-centre HD. Moreover, several German as well as international studies reveal an overall cost benefit of PD [17,18,19,20,21,22].

Thus, the question emerges as to why the PD rate in Germany is lower compared to other (wealthy) countries. While there might be some external determinants (e.g., patient-side and health care provider-side reasons) that account for the preferred use of HD in Germany, from the SHI perspective, differences in direct health care costs between the two dialysis modalities could be considered as a further reason [23]. To date, only few comparisons of the direct health care costs of HD and PD exist for the German health care system, and most of them are based on outdated data or did not control for important confounders, with none using propensity score-matching techniques [18,23,24,25].

Against this background and excluding societal costs, this study aims to compare PD and HD in terms of (i) direct health care costs and additionally regarding clinical outcomes such as (ii) hospitalisation rates as well as identification of predictors of hospitalisations and (iii) survival rates.

## 2. Materials and Methods

### 2.1. Study Design & Setting

Claims data of two health insurance funds of the German SHI were analysed in a retrospective cohort study (s. below). Patients were identified over a 4-year study period (2013–2016) using the ICD-10 diagnosis N18.5 for chronic kidney disease (CKD) and the German EBM (Einheitlicher Bewertungsmaßstab—Value Scale for outpatient services, s. below) codes for HD and PD. DAK-Gesundheit (Deutsche Angestellten Krankenkasse) and SBK (Siemens Betriebskrankenkasse) operate nationwide and cover together around 6.7 million insured [26,27]. Our data set contained all medical encounters of 28,553 patients from 2013 to 2016, identified by at least one documentation of ICD code N18.5 during this period (for identification of HD- and PD-patients, see below and Figure 1).

Based on this data set, we identified HD and PD patients and used propensity score matching (PSM) to minimise selection bias in the data and to create comparable and balanced HD and PD groups. For this purpose and to conduct our analyses, 2013 was taken as the baseline year and the years 2014–2016 were used as the observation period. 

### 2.2. Claims Data in the German SHI

In the German SHI the reimbursement of outpatient medical services is based on the EBM, a codebook which encodes all medical services that may be billed to the SHI [28]. Each EBM code is associated with a specific medical service. All medically indispensable services are reimbursed by the SHI and are therefore included in the claims data with EBM codes. This allows the use of claims data to analyse patient care and its impact, especially in the context of chronic diseases, such as many nephrological illnesses [1,29,30].

## 3. Study Population

### 3.1. Identification of Dialysis Modality via EBM Code

Patients with an age > 18 years and an ICD-10 diagnosis N18.5 for CKD were identified and classified as HD and PD patients depending on whether the EBM code of their dialysis was an in-centre HD (i.e., patients treated in an ambulatory dialysis centre by a team of nurses and medical assistants) or a home-based PD. Both incident and prevalent patients within the observation period were included, whereby they could not be distinguished.

We excluded the special dialysis modalities home hemodialysis (HHD) and intermittent peritoneal dialysis (IPD) via an additional EBM code, since both of them are rarely applied (see Appendix A, for the relevant EBM codes) [28].

### 3.2. Criteria for Inclusion of Participants

All patients who were identified as HD or PD patients using the above criteria for at least two consecutive quarters during the observation period 2014–2016 were included in our study population. This corresponds to a more specific version of the so-called m2q criterion, which is used for any patient with at least one identical ICD diagnosis in at least two quarters of a calendar year [31].

PD patients who were at the same time identified as HD patients based on the above-mentioned ICD codes were excluded from the study population. Similarly, HD patients who were also notified as being treated with PD were excluded, thus excluding all dialysis patients who switched treatment modalities during the time under evaluation. However, these switchers were taken into account in a sensitivity analysis. Four absolute contraindications (inflammation/diverticulitis of the bowel, reduced mental status, disorders of the peritoneum) for PD were determined (see Appendix A). If one of these diagnoses was coded within the baseline year 2013, the patient was assigned the corresponding contraindication. HD patients who had any of these contraindications were excluded as matching partners in the PSM. Patients who underwent a kidney transplant during the study period were also excluded.

## 4. Outcomes

### 4.1. Total Annual Health Care Costs

The primary endpoint was the total annual direct health care costs. These included costs of hospital treatment (inpatient services), costs of treatment in ambulatory care (outpatient services), costs due to sick pay, drug expenses, costs of therapeutic and medical aids, and the cost of rehabilitation. Because costs for transportation were not represented in the claims data of the two participating SHIs, these were calculated in a further analysis by one of the two SHI funds for the year 2017 using an extra dataset. Results were displayed in costs per year.

### 4.2. Hospitalisations

Hospitalisation rates for HD and PD patients were calculated by applying the Poisson test. A multiple Poisson regression model was used to identify predictors of hospitalisations. Age, sex, region of residence and the Charlson Comorbidity Index (CCI, see below) were included as independent covariates for both the HD and PD group separately.

### 4.3. Survival

Survival rates were calculated using Kaplan–Meier estimates. An observation was censored if the patient was still alive on 31 December 2016 or switched SHI before 31 December 2016. The log-rank test was applied to assess for statistically significant differences, and Kaplan–Meier curves were constructed to display the survival functions for HD and PD patients.

### 4.4. Charlson Comorbidity Index

The non-age-adjusted CCI is a measure of comorbid conditions and was calculated for all study patients using baseline diagnoses for the year 2013 [32,33]. The CCI considers 17 diagnoses, among them diabetes and congestive heart failure, each of which is assigned an integer weight from one to six, with higher scores indicating a higher burden of comorbidity. Finally, the summation of the weighted comorbidity scores yields the CCI (see Appendix A).

### 4.5. Statistical Analysis

All cases with missing values regarding the variables age, sex, region of residence (for definition see Appendix A) and CCI were excluded from further analyses. No imputation of missing values took place.

The statistical significance of differences in continuous variables was assessed by the two-tailed Wilcoxon rank-sum test. For count data, the Poisson test was used. A Chi-square test was applied to detect statistically significant differences among percentage values in different groups.

To account for general parameter uncertainty or skewness in the costs, the associated confidence intervals were calculated by bootstrapping.

A *p*-value < 0.05 indicated statistical significance. All analyses were conducted using IBM^®^ SPSS^®^ Statistics for Windows, version 26.0, Armonk, NY, USA; IBM Corp., MedCalc, version 19.5.3, and R, version 4.1.0 [34].

### 4.6. Propensity Score Matching

Randomised controlled clinical trials are considered the gold standard for evaluating the effectiveness of medical interventions. However, since randomisation is not possible in a retrospective cohort study, matching techniques such as PSM offer an alternative for generating comparable groups by adjusting for observed confounders. In our analysis, propensity scores were estimated using logistic regression with enrolment into the PD group (yes = 1/no = 0) as the outcome variable of the matching process. The propensity score (PS) attributed to an insurant is the individual probability of being in the PD group taking into account the observed covariates measured at the baseline, and can be used as a matching variable that replaces all independent covariates with a single scalar. The following independent covariates were included: age, sex, region of residence, and health status at baseline captured by the CCI. In the subsequent matching process, the following specifications were combined and tested to optimise the balance between the HD and the PD group: 1:n (1 ≤ *n* ≤ 5) ratios, PSM as well as combined PS and exact matching with the nearest neighbour as matching algorithm, logistic regression as estimation procedure and a caliper of 0.1. Patients without a matching partner were excluded from further analyses. Standardised differences of <10% were considered an acceptable degree of balance between the two groups for a given covariate [35,36,37,38].

### 4.7. Sensitivity Analysis

The robustness of the analysis was assessed by creating different PSM ratios and combining different matching techniques. For each of these models, a comparison of total direct health care costs between the two dialysis modalities was performed.

Because the data set did not include information about transportation costs, these were calculated by adding an internal analysis of one of the participating SHIs for the year 2017.

Total annual direct health care costs were calculated for switchers between modalities.

## 5. Results

### 5.1. Propensity Score Analysis and Study Population

Our data set contained 28,553 insurants (total cases) who had at least one documentation of ICD code N18.5 between 1 January 2013 and 31 December 2016. Among those, after the exclusion of switchers (*n* = 271, starting with HD: 88/PD: 183), a total of 9894 HD and PD patients (‘pure’ cases) were identified for the observation period from 1 January 2014 to 31 December 2016. Further exclusion of cases due to missing values or PD contraindications resulted in 6829 HD patients and 439 PD patients who constituted the basis for the PSM: The use of all 1:n (1 ≤ *n* ≤ 5) ratios and the different matching techniques mentioned above revealed that the best result in terms of balance (equal distribution of covariates) was obtained using a combined PS (on CCI) and exact matching (on age, sex, region) with a 1:1 ratio (HD vs. PD), nearest neighbour as matching algorithm and a caliper of 0.1. After matching, all standardised differences were below 10% (values not shown), and beforehand significant differences in age, region, and CCI disappeared, indicating that the matching adjusted the data appropriately. Finally, the PSM resulted in 436 HD and 436 PD patients (*n* = 872 patients, see Figure 1, Table 1).

### 5.2. Total Annual Health Care Costs

Table 2 displays the average total annual direct health care costs of the two dialysis modalities broken down into the different types of costs. The total annual costs were slightly higher for HD patients (mean: EUR 47,501) than for PD patients (mean: EUR 46,235), but the difference (mean: EUR 1266; 95% CI: −2879–5487) was not significant (*p* = 0.557).

Significant differences could be observed for different cost types, such as drug expenses (*p* = 0.003), sick pay (*p* = 0.048), therapeutic aids (*p* = 0.043), and rehabilitation (*p* = 0.017).

### 5.3. Sensitivity Analysis

A comparison of average total annual direct health care costs between the two dialysis modalities for each of the tested models revealed no significant differences. Additional consideration of transportation costs resulted in an annual cost advantage of EUR 7000 in favour of PD.

Total annual direct health care costs for switchers were EUR 48,774.75.

### 5.4. Hospitalisations

#### 5.4.1. Hospitalisation Rates

The patient years related to the study population amounted to 1.13 and 1.14 for HD patients and PD patients, respectively. Furthermore, there were 1.94 admissions in the PD group, with a rate of 1.72 admissions per patient-year and 15.24 hospital days per patient and year. In the HD group, there were 1.99 admissions, with a rate of 1.74 admissions per patient-year and 15.21 hospital days per patient and year. These differences were not significant (hospitalisation rates: *p* = 0.610, hospital days: *p* = 0.853).

#### 5.4.2. Predictors for Hospitalisations

In the Poisson model only the CCI turned out to be a predictor of hospitalisations in the HD group, i.e., a higher degree of comorbidity resulted in a higher risk of hospitalisation (beta = 0.072, *p* = 0.000). In the PD group, the risk of hospitalisation was similarly increased (beta = 0.076, *p* = 0.000), and women had a significantly lower risk of hospitalisation (see Appendix A).

### 5.5. Survival Analysis

The proportion of censored cases in the study population was 76.3%. A total of 95 (21.8%) patients died in the HD group, and 112 (25.7%) patients died in the PD group (Table 3). Since the observation period covered three years, the maximum survival time amounted to 1095 days (patients at risk at this time point: HD 322/PD 308). During the observation period, the average 3-year survival was 963 days in the PD group, whereas in the HD group it was 978 days (see Table 3). Thus, patients from the HD group lived an average of 15 days longer than PD patients, whereby this difference was not significant. 

According to the curves of the Kaplan–Meier estimate, the survival rate over the observation period was slightly lower in the PD group than in the HD group (see Figure 2). However, the difference in survival curves was not significant (Mantel-Cox log-rank test: *p* = 0.207).

For comparison with the unmatched dataset see Appendix A.

## 6. Discussion

In this study, we compared the total direct annual health care costs, hospitalisation rates and survival rates between the dialysis modalities HD and PD. The comparison of these costs (excluding transportation costs) revealed a tendency towards higher costs for HD patients, although the difference was not significant. However, we found significant differences in various cost types, such as drug expenses, which were significantly higher for HD patients than for PD patients. We deliberately excluded patients who switched from one dialysis modality to another to sharpen our analysis. This did not lead to a bias, as a sensitivity analysis revealed that the total direct annual health care costs of switchers hardly differed from those of HD and PD patients.

Since transportation costs were not included in the SHI data models, we could not calculate these costs in the primary analysis. However, an internal analysis (for the year 2017), which could be conducted by one of the two participating SHIs, showed that annual transportation costs were on average EUR 7000 higher for HD patients compared to PD patients. This is in line with other German studies that identified comparable cost advantages for PD over HD [17,18]. Furthermore, this result is supported by a cost comparison carried out in the Coreth study (2015), which showed approximately EUR 11,000 higher annual transportation costs per patient for HD [39]. Finally, another analysis showed that the transportation cost advantage of PD amounted to >€14,000 per patient and year [40]. However, this calculation is based on the average distance of patients from their home to the dialysis centre and the associated mean fares.

Based on these findings, annual transportation cost of about EUR 7000 per patient in favour of PD in our study appears to be a conservative estimate. Given this and the 95% CI for the mean difference in total direct costs (−2879–5487), the potential cost benefit of PD in Germany can be expected between EUR 4100 and EUR 12,500.

All the above studies showed an overall cost advantage for PD, which was also confirmed by several international studies. According to a Taiwanese cohort study which also used matching techniques, the average lifetime costs per patient are higher for HD than for PD (USD 237,795 vs. USD 204,442) [19]. A cost comparison from Panama for 2015 resulted in lower total annual costs for PD than for HD [20]. The question arises of what cost savings would be possible with an increased utilisation of PD. Treharne et al. reported in their analysis that the PD rate in the UK is currently at 22%. Compared to this, PD rates of 30% or 50% would result in cost savings of GBP 3180 or GBP 5238 per patient, respectively, in a five-year scenario. In a 10-year scenario cost savings would be GBP 4102 or GBP 6758, respectively [21]. A US study states that raising the PD rate from the current 8% to 15% could potentially save US health insurers over USD 1.1 billion over 5 years. In contrast, dropping the PD rate to 5% could cost an additional USD 401 million over this period [22].

Regarding hospitalisation rates we could not detect any significant differences between HD and PD, which is in line with another study [41]. Contrary to the assumption that the frequency and duration of hospitalisation due to serious infectious complications (peritonitis) is likely to be higher in peritoneal dialysis patients than in HD patients [42], there was no difference in these variables between PD and HD patients in our analysis. This is in line with the results of the North Thames study, which also showed no difference in hospitalisation frequency and duration between elderly PD and HD patients [43]. It is possible that the procedure-associated complications requiring inpatient treatment will be different between the two groups, but comparable in terms of frequency and duration of hospitalisation. In terms of predictors of hospitalisations, in the HD group, only the covariate CCI emerged as a predictor, while in the PD group, women had a significantly lower risk of hospitalisation than men.

Finally, our survival analysis showed that patients undergoing PD did not have a significantly different overall survival rate compared to HD patients. This finding is concordant with other studies [4,44].

The outcomes in terms of hospitalisations and survival time support the validity of the cost comparison regarding bias of selection. Therefore, it is all the more remarkable that also our analysis confirms that PD is used far less frequently than HD as dialysis modality, especially in light of the fact that both are considered medically equivalent, and that PD allows for the preservation of autonomy [10].

### 6.1. Strengths and Limitations

Our analysis has several strengths. It is based on a large database that allowed us to extract 6829 HD and 439 PD patients from 2014–2016 and use them for the matching. Thus, the PD proportion in our sample was comparable to the one stated by the official national report for 2016 [45]. A further strength of our analysis is that we could additionally consider and analyse clinical aspects such as hospitalisations and survival. Finally, we used combined PS and exact matching to minimise treatment selection bias from socioeconomic variables and health status.

However, our work has some limitations. Regarding the data basis, some potentially relevant baseline variables were not available. These included, e.g., duration of disease or income. In addition, incident and prevalent cases could not be distinguished during the observation period because it was not possible to trace back far enough, disallowing for controlling health care costs at baseline. This concerns hospitalisations as well.

Regarding our analysis, the used matching techniques generally balance only measured baseline variables between subjects, whereas randomisation would provide a comprehensive structural balance between treatments arms. Therefore, for matched subjects, there might remain imbalances based on unmeasured characteristics, as mentioned before. Thus, generalisation of the findings to all ESRD patients is restricted.

In general, research based on claims data is associated with some inherent limitations because such data only consists of procedures relevant to reimbursement. Parts of treatment that are not reimbursed were not included. Another limitation is that transportation costs could not be included in the primary analysis and could only be estimated based on an internal analysis of one of the two SHIs beyond the observation period.

### 6.2. Future Research

Since there are only few comparative cost analyses from Germany regarding HD and PD and transportation costs may have a major impact, future studies should pay special attention to them.

## 7. Conclusions

In our study no significant differences between the two dialysis modalities HD and PD regarding total annual health care costs and clinical outcomes could be identified. On the other hand, since transportation costs can be estimated to be significantly higher for HD than for PD, there may be a total cost advantage for PD, which suggests a preferential reimbursement of this modality.

Hence, if transportation costs are taken into account, PD seems to be a cost-saving alternative for the SHI for the treatment of ESRD. PD may enable patients to integrate dialysis into their daily life and may help suitable patients to maintain their autonomy. Patients should be educated about the possibility of PD and suitable patients should be educated to be able to perform PD.

## Figures and Tables

**Figure 1 ijerph-19-14007-f001:**
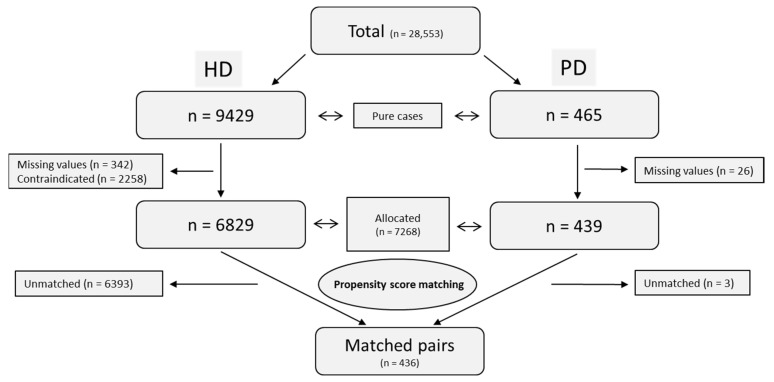
Flowchart study population.

**Figure 2 ijerph-19-14007-f002:**
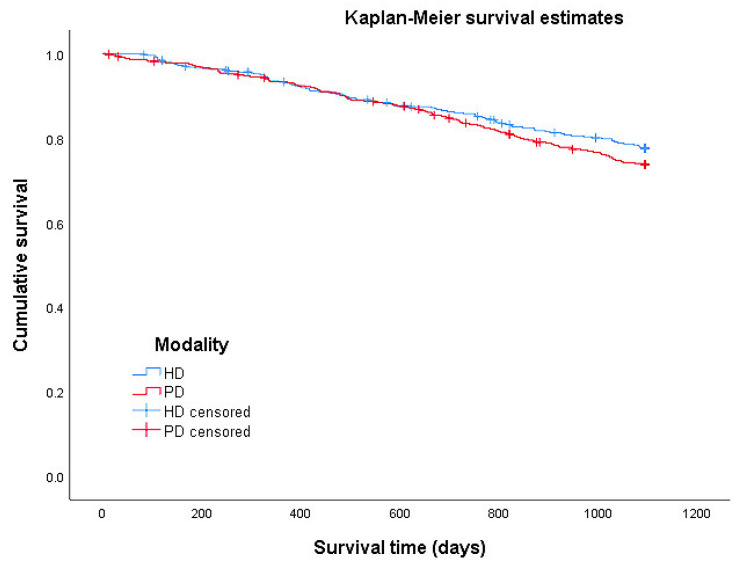
Overall survival (HD: *n* = 436, PD: *n* = 436, log-rank = 0.207).

**Table 1 ijerph-19-14007-t001:** Characteristics before and after matching.

		Before Matching					After Matching				
		HD (*n* = 6829)		PD (*n* = 439)			HD (*n* = 436)		PD (*n* = 436)		
		*n*	%	*n*	%	*p*-Value	*n*	%	*n*	%	*p*-Value
Sex	Women	2860	41.9	181	41.2	0.789	178	40.8	178	40.8	1.000
	Men	3969	58.1	258	58.8		258	59.2	258	59.2	
Age	18–44 years	392	5.7	71	16.2	0.000	68	15.6	68	15.6	1.000
	45–64 years	1850	27.1	197	44.9		197	45.2	197	45.2	
	65–74 years	2032	29.8	108	24.6		108	24.8	108	24.8	
	74–84 years	2041	29.9	57	13.0		57	13.1	57	13.1	
	≥85 years	514	7.5	6	1.4		6	1.4	6	1.4	
Region	North	1498	21.9	66	15.0	0.001	65	14.9	65	14.9	1.000
	East	1105	16.2	65	14.8		64	14.7	64	14.7	
	South	2019	29.6	137	31.2		137	31.4	137	31.4	
	West	2207	32.3	171	39.0		170	39.0	170	39.0	
Age Mean (SD)		68.5 (13.2)		59.3 (14.4)		0.000	59.8 (14.2)		59.4 (14.6)		0.747
CCI Mean (SD)		6.7 (3.5)		5.4 (3.3)		0.000	5.4 (3.2)		5.4 (3.3)		1.000

**Table 2 ijerph-19-14007-t002:** Annual costs per patient (in EUR) according to type of dialysis (HD or PD) (HD: *n* = 436; PD: *n* = 436).

Cost Types	HD	PD	HD-PD ^a^	95% CI *	*p*-Value
Total	47,501	46,235	1266	−2879–5487	0.557
Outpatient services ^1^	24,158	23,335	823	−864–2413	0.336
Drug expenses	8144	5961	2183	736–3614	0.003
Sick pay	304	575	−271	−544–−3	0.048
Therapeutic aids	602	423	179	2–341	0.043
Inpatient services ^1^	13,974	15,799	−1825	−5545–1832	0.317
Rehabilitation	319	143	176	36–328	0.017

* Confidence intervals were calculated by bootstrapping (*n* = 10,000); ^a^ Difference of HD and PD. For comparison with the unmatched dataset see Appendix A; ^1^ Including costs for dialysis therapy.

**Table 3 ijerph-19-14007-t003:** Survival data.

	Total	Event		Censored		Survival in Days		
		N	%	N	%	Estimator	SE	95% CI
HD	436	95	21.8	341	78.2	977.5	12.6	952.8–1002.2
PD	436	112	25.7	324	74.3	963.3	13.0	937.9–988.7
Total	872	207	23.7	665	76.3	970.4	9.0	952.7–988.1

## Data Availability

The MAU-PD study (Multidimensional analysis of causes for the low prevalence of ambulatory peritoneal dialysis in Germany) is registered at the German Clinical Trials Register (DRKS-ID: DRKS00012555).

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
