# Peer review of "Hemodialysis and Peritoneal Dialysis in Germany from a Health Economic View—A Propensity Score Matched Analysis"

_ijerph, 2022, doi:10.3390/ijerph192114007_

Round 1

Reviewer 1 Report

Overall, it is a well written paper, with methodology being sufficiently described. 

However, some of the limitations were not fully addressed. 

First, the clinical characteristics of ESRD patients were not fully identified. For example, characteristics related to severity of disease which is definitely a driving factor of the subsequent health care cost were not fully explored using the claims data. To list some of them: duration of disease, therapeutic modalities or treatments along the disease course, disease related hospitalizations in the baseline year, even the disease related health care cost in the baseline year, etc. Those are the important factors to be included in the PSM logistic regression model to better inform the PS score. 

Second, by looking at the PSM before and after covariates, it was quite evident that PD group were in general younger and healthier than HD group. So by applying PSM, the final analysis only compared HD and PD strictly in a subset of ESRD patients which may represent the younger and healthier ESRD patients. It is quite questionable to generalize the study findings to overall ESRD patients. 

Third, there are other factors in the observation years that may contribute to the choice of PD vs HD. For example, a patient being with his/her family or caregivers might use PD more. A patient being in workforce might use PD more. Those who were in higher social economic status (eg, higher income) might use HD more. The region variable only has four categories which represent a too large geographic area. A suggestion is to look at much finer/smaller geographic region that might indicate the social and economic differences across regions. The audience would love to see if these factors were explored using the claims data.   

Author Response

Thank you very much for your helpful comments! Below you find our corresponding answers.

“First, the clinical characteristics of ESRD patients were not fully identified. For example, characteristics related to severity of disease which is definitely a driving factor of the subsequent health care cost were not fully explored using the claims data. To list some of them: duration of disease, therapeutic modalities or treatments along the disease course, disease related hospitalizations in the baseline year, even the disease related health care cost in the baseline year, etc. Those are the important factors to be included in the PSM logistic regression model to better inform the PS score.”

Response: Thank you for this important comment. We partially agree with you on this point. It is true that the quality of the analysis could be increased by adding these variables. However, we think we could partially compensate this limitation by including the Charlson Comorbidity Index. We completed the manuscript accordingly (s. line 338-352).
Unfortunately, the variables available concerning patient characteristics were limited due to legal obstacles.
Ffurthermore, we were unable to determine the duration of disease, since we could not distinguish incident and prevalent cases in the data due to the limited observation period.
We described the dialysis related therapeutic modalities or treatments along the disease course by the PD and HD modalities or indirectly by the Charlson Comorbidity Index, respectively.
Since we did not have a controlled trial design with clearly identifiable incidence cases at baseline we could not control for hospitalisation rates and health care costs (which are at the same time the outcome) at baseline or include them in the PSM. Otherwise, we would have erroneously eliminated pre-existing differences between PD and HD when adjusting health care costs at baseline.

“Second, by looking at the PSM before and after covariates, it was quite evident that PD group were in general younger and healthier than HD group. So by applying PSM, the final analysis only compared HD and PD strictly in a subset of ESRD patients which may represent the younger and healthier ESRD patients. It is quite questionable to generalize the study findings to overall ESRD patients.”

Response: Application of PSM creates a subset of the total sample generating comparable groups disregarding those cases that are excluded by the PSM. However, the results shown in the supplementary material indicate that the health care costs are not significantly different even in the unmatched comparison. We adjusted the limitations accordingly (s. line 338-352).

“Third, there are other factors in the observation years that may contribute to the choice of PD vs HD. For example, a patient being with his/her family or caregivers might use PD more. A patient being in workforce might use PD more. Those who were in higher social economic status (eg, higher income) might use HD more.”

Response: We share your opinion, but as mentioned above, these variables are not or only incompletely available and thus can not be used for the PSM. We added this to the manuscript accordingly (s. line 338-352).

“The region variable only has four categories which represent a too large geographic area. A suggestion is to look at much finer/smaller geographic region that might indicate the social and economic differences across regions. The audience would love to see if these factors were explored using the claims data.”

Response: We agree with your point. Nevertheless, for data privacy reasons the postcodes are not provided in full, which inhibits a more detailed look at the geographic regions in Germany.

Reviewer 2 Report

The article is generally well written and the empirical part is sounding. I think this article is ready for publication after some minor adjustments. My suggestion mainly concerns the introduction and discussion:

1. It would be better to have more background about PD rates. The authors currently compared Germany with Hong Kong, Sweden, and Luxemburg. However, there might be institutional and socioeconomic differences among these places that caused the varying PD rates, which the authros did not go into detail. It would beneficial if the authors could articulate a bit about prior research and findings in these regions.

2. Are Hong Kong, Sweden, and Luxemburg the only places out there where studies have touched upon PD rates? I see the authors talking about many studies in other countries in the discussion. It seems to me that the article would read logically better if those studies could be mentioned in the introduction as well. 

3. This is related to the above point - I am currently not seeing the significance of this research because of the lack of a literature review in the introduction. I understand that studying PD rates in Germany is important. But hows do your scope and results contribute to our overall understanding of PD rates? This needs to be stated, 

4. You mentioned SHI and its potential relationship with PD rates in Germany. It would be good to have more information about that - is SHI means-tested for all patients? What number are you talking about by stating "highly remunerated"?

5. The current form of discussion misses explanations of your findings, which underplays your discussion to a section similar to the results. For example, you mentioned the insignificant relationship between hospitalization rates and HD and PD, but you did not explain your thoughts about why that is the case. Readers can understand the results by looking at your tables. But it is also important for the authors to make a stance/proposition in the discussion, and to give logically-sound explanations for the results. Although this is not a mechanism study, the article will benefit from your input about why nevertheless. 

Author Response

Thank you very much for your helpful comments! Below you find our corresponding answers.

1. It would be better to have more background about PD rates. The authors currently compared Germany with Hong Kong, Sweden, and Luxemburg. However, there might be institutional and socioeconomic differences among these places that caused the varying PD rates, which the authors did not go into detail. It would beneficial if the authors could articulate a bit about prior research and findings in these regions.

Response: We agree with your point and adapted the manuscript accordingly. We added PD rates and backgrounds (s. line 51-64).

2. Are Hong Kong, Sweden, and Luxemburg the only places out there where studies have touched upon PD rates? I see the authors talking about many studies in other countries in the discussion. It seems to me that the article would read logically better if those studies could be mentioned in the introduction as well.

Response: We share your opinion and adapted the manuscript accordingly (s. line 51-64).

3. This is related to the above point - I am currently not seeing the significance of this research because of the lack of a literature review in the introduction. I understand that studying PD rates in Germany is important. But how do your scope and results contribute to our overall understanding of PD rates?

Response: The focus of our study is not on PD rates, but on comparing the health care costs of both dialysis modalities and whether possible differences could be a reason why the PD rate in Germany is so low, while it is much higher in some other (wealthy) countries. We have précised this point in the introduction (s. line 51-64).

4. You mentioned SHI and its potential relationship with PD rates in Germany. It would be good to have more information about that - is SHI means-tested for all patients? What number are you talking about by stating "highly remunerated"?

Response: All patients in our sample are insured in the mentioned SHI funds. Therefore, we could not apply a statistical test for mean differences. We deleted the word "highly". The sentence means that there is no relevant difference in the amount of remuneration on the part of the SHI funds. In other words, the costs for both dialysis modalities are reimbursed equally by the SHI.

5. The current form of discussion misses explanations of your findings, which underplays your discussion to a section similar to the results. For example, you mentioned the insignificant relationship between hospitalization rates and HD and PD, but you did not explain your thoughts about why that is the case. Readers can understand the results by looking at your tables. But it is also important for the authors to make a stance/proposition in the discussion, and to give logically-sound explanations for the results. Although this is not a mechanism study, the article will benefit from your input about why nevertheless.

Response: We have taken up your valuable suggestion regarding hospitalisations in the discussion (s. line 304-313).